# Performance Evaluation of Lateritic Subgrade Soil Treated with Lime and Coir Fibre-Activated Carbon

Sakina Tamassoki [1,2], Nik Norsyahariati Nik Daud [1,3,*], Fauzan Mohd Jakarni [1], Faradiella Mohd Kusin [4], Ahmad Safuan A. Rashid [5] and Mohammad Jawed Roshan [2,5]

1. Department of Civil Engineering, Faculty of Engineering, Universiti Putra Malaysia, Serdang 43400, Selangor, Malaysia
2. Faculty of Transportation Engineering, Kabul Polytechnic University, Kabul 1001, Afghanistan
3. Housing Research Centre (HRC), Faculty of Engineering, Universiti Putra Malaysia, Serdang 43400, Selangor, Malaysia
4. Department of Environment, Faculty of Forestry and Environment, Universiti Putra Malaysia, Serdang 43400, Selangor, Malaysia
5. Department of Geotechnics and Transportation, Universiti Teknologi Malaysia, Johor Bahru 81310, Johor, Malaysia
* Correspondence: niknor@upm.edu.my

**Abstract:** The subgrade layer's stability considerably influences the long-term performance of pavement systems. This study investigates the influence of lime as a traditional stabiliser and activated carbon with coir fibre (ACF) as waste materials and an environmentally friendly binder to stabilise lateritic subgrade soil. Experiments, including the one-dimensional consolidation and unconfined compressive strength (UCS) tests, have been conducted to investigate the geotechnical properties of stabilised soil in various percentages of additives 3%, 6%, 9%, and 12% lime and 1%, 2%, and 3% ACF. The results demonstrate that 3% ACF and 12% lime can significantly improve the strength parameters and decrease the void ratio and permeability in the stabilised soil. Furthermore, microstructural analysis was performed before and after stabilisation for optimum content. The microstructural analysis proves that AC and lime particles fill soil voids, and gel formation binds the soil particles in the stabilised soil matrix. The results show that 3% ACF stabilised soil is comparable with 12% lime in UCS value and decreasing void ratio. Furthermore, both are suitable for subgrade of low-volume road stability according to Malaysian standards.

**Keywords:** lateritic soil; activated carbon; fibre; lime; consolidation; compressive strength; microstructure

## 1. Introduction

Ever increasing population and scarcity of suitable land lead to passing the transportation alignment through regions with marginal soils. However, the marginal soils possess low engineering and mechanical properties. Moreover, the rainfall infiltration further decreases the stability of transportation infrastructures [1]. Therefore, it needs to be stabilised before being used for infrastructure construction [2]. To date, various stabilisation such as chemical stabilisation [3–6], reinforcement [7–9], deep mixing [10,11], and prefabricated vertical drain [12,13] have been utilised to improve the engineering properties of unsuitable soils, and this issue leads to stable constructions.

Soil stabilisation by chemical admixtures has recently gained popularity in geotechnical engineering. Increased modulus of elasticity and resilient modulus, strength properties, reduced plasticity index, permeability, decreased swelling potential and volume instability, deformation and settlement, and improved durability are only a few of the benefits of lime as stabilisation materials [14,15]. In places with wetting and drying cycles, utilising low lime concentration to enhance the mechanical properties of soil is not an effective strategy since the swelling potential of soil is not greatly diminished [16]. On the other

hand, adding lime to expansive clay soils, such as black cotton soil, is beneficial because the flexibility of the soil is reduced when lime is added. The application of lime as a stabiliser agent is not helpful for some cases where soil bearing capacity, density, and hydraulic conductivity are insufficient [17]. Durability, compressibility, and soil strength are increased by lime stabilisers while having varying effects on permeability. Moreover, sulfate attack, carbonation, and environmental effect are disadvantages of using lime as soil stabilisation.

Adding fibre to lime-treated soil can reduce the brittleness of treated soil [18]. As a reinforcement, synthetic and natural fibre increases tensile strength, improves stability, and reduces soil's lateral deformation [19]. Furthermore, fibre minimises the risk of lime stabilisation causing brittle failure. In soil stabilisation with cement, including polypropylene fibres, increased tensile strength, density, and initial elastic modulus. Fibre added to a cement-stabilised clay soil enhanced tensile strength, and reduced swelling potential, shrinkage, and crack width [20]. Moreover, fibre positively influenced the flexural behaviour of cement-based soil stabilisation [21].

In sandy soil, discarded tyre textile fibres were used to improve damping ratio, resilient modulus, and permanent strain [22]. Fibre content and aspect ratio reduced critical confinement stress and enhanced shear strength in sandy soil [23]. Moreover, using fibre in embankments improved slope safety factor, strength, and stability [24]. Furthermore, when fibre-reinforced soil was exposed to freeze-thaw cycles, Scanning Electron Microscopy (SEM) pictures revealed that the fibres remained intact despite the repeated freeze-thaw [25].

Several studies have assessed the impact of including natural fibres in stabilised soil mass with additives. This previous research has revealed that adding natural fibre reinforcing to the soil led to a significant enhancement in strength and a decline in soil stiffness. Table 1 provides a summary of the components explored in prior investigations.

**Table 1.** Natural fibre and additives investigated in some previous research.

| Test | Soil | Stabilising Admixture | Fibre Reinforcing | Days Curing | Ref. |
|---|---|---|---|---|---|
| UCS and DST | SC (Marginal soil) | Cement and Fly Ash | randomly distributing 2% coir fibre | 3 | [26] |
| USC | Clay (A kaolin type) | Lime | 0.75% basalt | 90 | [27] |
| UCS | CL | Lime and fly ash | 0.5% sisal fibre | 7 | [28] |
| TPB, TST | marine clay soil | Lime and cement | 1% coconut fibre | 28 | [29] |
| UUT | Hefei clayey soil | Lime | 0.4 wheat straw | 28 | [30] |
| UCS | CH | Lime | 0.5% coir fibre | 7 | [31] |
| TPB, cyclic loadings, | A-2-6 A-7-5 | Cement | 0.15% hemp fibre | 7 | [32] |
| UCS, STS, | SP | Cement | Kenaf fibre | 28 | [33] |

Direct shear test (DST), indirect tensile strength (ITS), unconsolidated–undrained triaxial (UUT), unconfined compressive strength (UCS), three-point bending (TPB), splitting tensile strength (STS).

Coir fibre is a natural fibre and one of the productions of coconut waste. Coconut waste is one of the most abundant waste products in agriculture, with a global production of 62.5 million tonnes annually in over 90 countries worldwide [34,35]. Coir fibre has complex properties; each layer structure includes two layers; the first is called the thin primary wall, and the second is called the secondary wall. The mechanical properties of the fibre can be determined in the thick middle layer of this secondary wall, which consists of three layers [36]. Coir fibre has a high lignin content but a low cellulose content, making them extremely strong, resilient, and durable. It has a high resistance to abrasion, fungal and bacterial decay, and pilling. Furthermore, coir fibres can tolerate months of soaking without deterioration [37].

Activated carbon (AC) is another material that improves soil's compression strength, shear strength [38] and California bearing ratio (CBR) value [39]. AC contains carbonaceous material and has been derived from a wide variety of carbon-rich raw materials, including

discarded apple pulp, coconut shells, straw, sugarcane bagasse, apricot stone shells, coir pith, sawdust, peanut husk, and olive stones [40]. It is a widely used method in the environmental sector for removing pollutants from air or water streams in industrial and municipal processes such as groundwater remediation, wastewater treatment plants, oil removal, and air purification by adsorption [41]. AC has a large surface area to absorb $CO_2$ and contaminants in submicroscopic pores. Moreover, it is stable in acidic and basic situations [42,43].

This study investigates the effect of lime as a traditional calcium-based stabiliser and coir fibre with activated carbon obtained from waste agriculture materials on the mechanical properties of lateritic soil. First, pH tests are conducted to determine the initial lime consumption. After that, UCS tests are conducted to find the influence of various lime percentages and activated carbon with and without coir fibre on the mechanical properties of stabilised specimens. Then, the consolidation test is performed for optimum content to determine the void ratio and permeability. Finally, energy dispersive X-ray analysis (EDX), surface area analysis (BET) and field emission scanning electron microscopy (FESEM) tests are performed to show the microstructure of untreated and treated soil to better understand the stabilised soil reinforcement mechanism.

## 2. Laboratory Investigations

### 2.1. Materials

Lateritic soils are generally unsuitable and weak for the construction of infrastructure facilities. This study collected soil at Universiti Teknologi Malaysia, Johor campus. The soil is categorised as A-7-5 [44]. The chemical components of the lateritic soil are shown in Table 2. Moreover, Figure 1 depicts the particle size distribution of lime, AC, and soil. The soil distribution is within the ranges discovered in previous studies [45].

**Table 2.** Chemical composition of soil and additives.

| Composition | (%) by Weight | | |
|---|---|---|---|
| | Soil | Activated Carbon | Lime |
| $Fe_2O_3$ | 57.57% | 11.27 | - |
| $SiO_2$ | 20.85% | - | - |
| $Al_2O_3$ | 19.507 | - | - |
| $K_2O$ | 1.72% | 17.68 | 0.30 |
| MnO | 0.15% | 2.57 | - |
| $Cr_2O_3$ | 0.14% | - | - |
| $As_2O_3$ | 0.05% | - | - |
| CaO | - | 39.77 | 98.85 |
| $P_2O_5$ | - | 16.62 | - |
| $SO_3$ | - | 7.83 | 0.80 |
| ZnO | - | 3.47 | - |
| CuO | - | 0.62 | - |
| SrO | - | 0.18 | 0.05 |

Regarding Figure 1, laser diffraction (LD) tests were conducted for lime, AC, and soil particles less than 75 microns. While sieving was used for soil particles more than 75 microns. Due to time savings and high accuracy, this study combined laser diffraction with conventional techniques [46]. Moreover, laser diffraction is more precise in evaluating the fine contents of residual soil [47], and hence it was employed instead of the hydrometer method.

In Table 2 chemical composition of additive materials is presented. AC coconut derivative was obtained from Evachem company, and lime was supplied by Lhoist company in Malaysia. AC is a kind of carbon that filters organic pollutants from the air, water, and other applications [41]. AC enhances the surface area for chemical reactions due to low-volume porosity [48]. The significant oxides of AC and lime were achieved by

X-ray Fluorescence (XRF) testing. About 40% of AC and 98.85% of lime are contained calcium oxide.

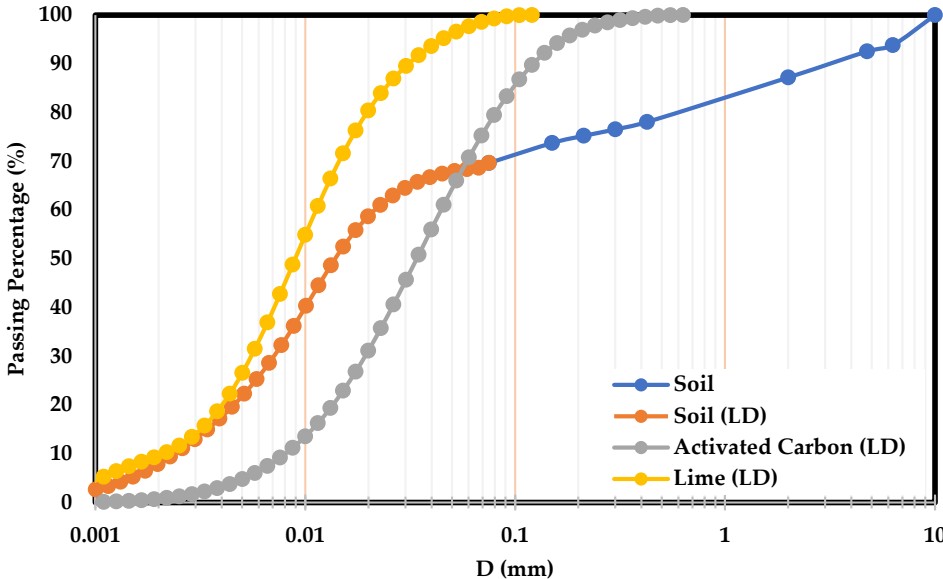

**Figure 1.** Particle size distribution of lime, AC, and soil.

Moreover, Coir fibre as reinforcement besides AC is used in this research. It is a waste and environmentally friendly material from a coconut's husk and is a fibrous material, cheap, and locally available. Coir fibre has higher tensile strength, is lighter, contains more hemicellulose and lignin, and degrades more slowly than other natural fibres. The coir fibre in this study has an average tensile strength of 125 MPa. Figure 2 presented an image of lateritic soil, lime, activated carbon, and coir fibre.

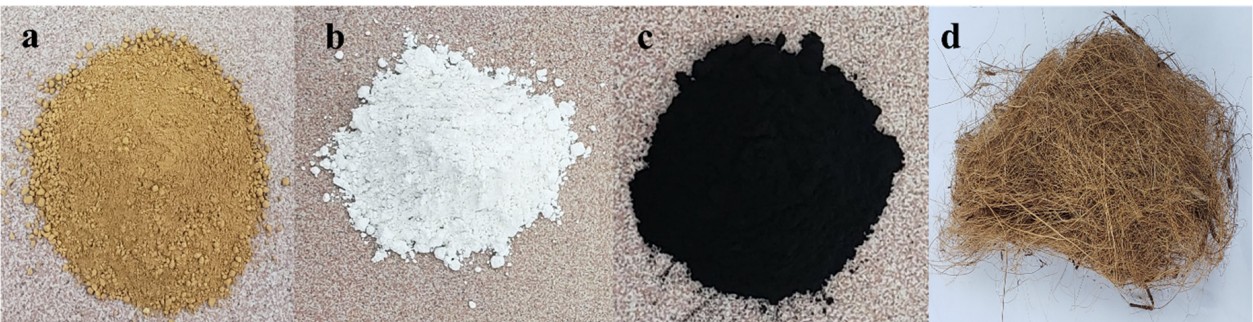

**Figure 2.** Image of (**a**) soil, (**b**) lime, (**c**) activated carbon, (**d**) coir fibre.

### 2.2. pH Test

The initial lime consumption was determined according to British Standard (BS) 1377: Part 3: 1990 [49]. The pH test indicates the lowest amount of lime that must be added to soil to cause a substantial change in characteristics. The very alkaline soil pH value (12.4) facilitates the dissolving of aluminous and siliceous chemicals from the lattice of clay minerals during lime stabilisation processes. Calcium silicate hydrate (CSH) and calcium aluminium hydrate (CAH) gels are formed when compounds released from the lattice of clay minerals combine with calcium ions in pore water, coating and bonding soil particles. The development of cementitious pozzolanic reactions is ensured by stabilising clayey soil with lime amounts more than the initial consumption of lime value [50].

### 2.3. Unconfined Compressive Strength Test

The mechanical characteristics of stabilised specimens are investigated using unconfined compression tests. The UCS testing is performed according to BS1377: Part 7:1990 [49] and a 1 mm/min rate to determine stabilised soil's compressive strength ($q_u$). The specimens are compacted, similar to the compaction test in three layers.

### 2.4. One-Dimensional Consolidation Test

The one-dimensional consolidation settlement tests were performed according to British Standard BS1377: Part 5:1990 [49] to investigate the consolidation behaviour of treated soil. Consolidation settlement is the vertical displacement that occurs in a soil specimen due to water expulsion from the voids resulting in volume reduction. All specimens at their optimum moisture content were compressed statically inside the consolidation ring in three equal layers to obtain maximum dry density [51] and then cured. All samples were subjected to vertical stresses of 50, 100, 200, 400, and 800 kPa. Each load increment was kept for 24 h to achieve 90% of consolidation before taking the final reading or increasing the load. Unloading readings have been taken at 200 and 50 kPa.

### 2.5. Microstructural Analysis

In this study, EDX and FESEM analyses have been performed on natural and modified soil to explore the microstructural changes of stabilised soil based on the previous study's procedure [52]. These tests have been conducted using a Hitachi SU8010 machine. UCS samples are used to obtain specimens for microstructural testing. Before FESEM analysis, a solid and tiny specimen is covered with platinum using a vacuum sputtering coat. Then, quantitative analyses of natural and treated lateritic soil are conducted by the EDX experiment [53].

### 2.6. Surface Area Analysis (BET)

The BET test has been conducted to evaluate changes in pore size distribution and surface area of natural and modified specimens in this research. Since most chemical reactions in soils occur at the surface area of particles, it is a parameter in investigating how the soil comes into contact with its surroundings chemically and physically [54]. This technique collects inert gas adsorption isotherm data and models it using the BET isotherm equation [55]. This is one of the most widely used techniques for quantifying external pore size distribution and surface area [56].

## 3. Test Results and Discussion

### 3.1. pH

Figure 3 indicates plots of pH values versus days for various lime and activated carbon contents. The pH of the investigated lateritic soil is 4.05, which indicates that it is acidic. The pH value of 12% lime-treated soil is 12.42, but it drops to 11.13 after more than 150 days. In this study, the results for pH values with rising lime and curing time were consistent with [57] for cement and [58] for lime, in that the authors reported that pH declined during curing time. Likewise, the pH decreased by 3%, 6%, 9%, and 12% Lime during curing time in the current study. At 12% lime, the alumina and silica components in the soil are dissolved out of the soil, making it accessible to combine with $Ca^{2+}$ to develop calcium silicates and aluminates as cementing products. Based on the pH test results, 12% lime is the initial lime consumption. When a soil has a pH value smaller, it needs more lime content to improve soil strength [57] due to significant hydration products formed (such as CAH, CSH, and hydrated lime). Lime stabilisation decreases volume change and Atterberg limits of soils but increases the strength and shrinkage limit of the soil-cement matrix [59]. However, in this study, the four percentages of the lime were employed (3%, 6%, 9%, and 12%) to treat lateritic soil. AC also increased pH to 6.6, 6.8, and 7 for 1%, 2%, and 3% AC, respectively. It is evident that 3% lime increases pH value significantly in comparison with AC because lime is an alkali material, while AC is neutral in this study.

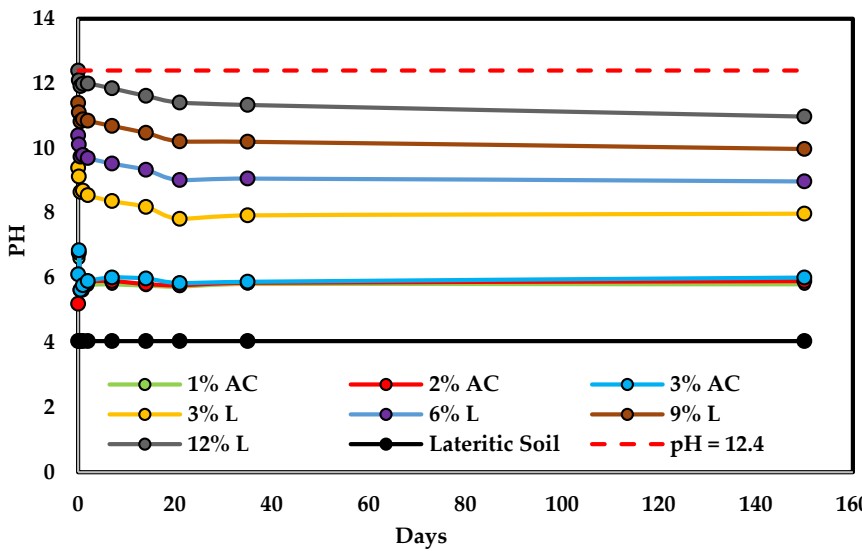
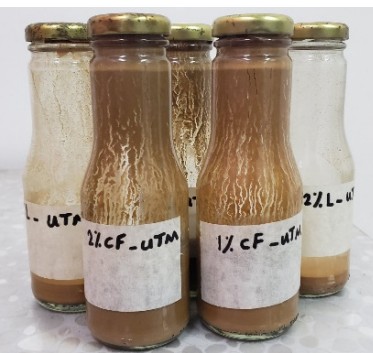

**Figure 3.** Variation in pH value after adding the various percentage of AC and lime.

### 3.2. Unconfined Compression Strength

Typical failure patterns of untreated and treated soil specimens with AC and lime are shown in Figure 4. The untreated sample has a bulging shape in the failure plane, while treated specimens were sheared in an inclined plane. The findings of [45] are consistent with the failure plane pattern observations of this study.

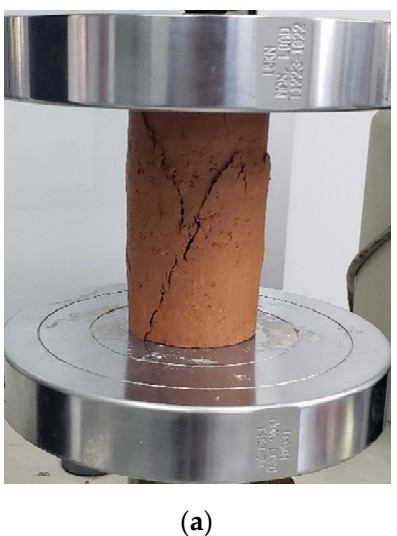
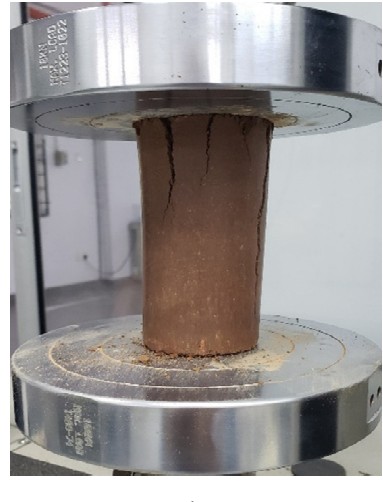
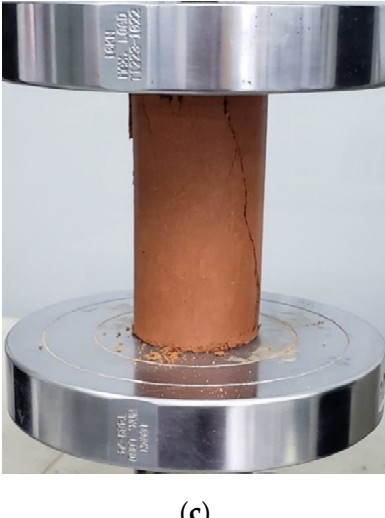

| (a) | (b) | (c) |

**Figure 4.** Failure patterns of UCS test: (**a**) lateritic soil, (**b**) AC-treated soil, and (**c**) lime-treated soil.

The compressive strength of AC and lime specimens is illustrated in Figures 5 and 6. First, the soil was examined for 1%, 2%, and 3% AC content. Then, the soil was examined for 1%, 2%, and 3% AC, and just 0.5% fibre due to using more than 5% decreases the compacted density, reducing soil strength [60]. Moreover, the soil was stabilised with 3%, 6%, 9%, and 12% lime under two and four weeks of curing.

The soil compressive strength was raised with rising AC and ACF content. For instance, the un stabilised UCS value of 200.87 kPa enhanced to 243.65 kPa, 306.31 kPa, and 545.40 kPa for 1%, 2%, and 3% AC. Moreover, the 0.5% coir fibre combination further improved the UCS of AC-treated soil, as shown in Figure 5. Likewise, Crane et al. [61] found that adding powder-activated carbon enhances soil's UCS value; some previous

research demonstrated that coir fibre enhanced clay soil's bearing capacity, stiffness, and strength [62].

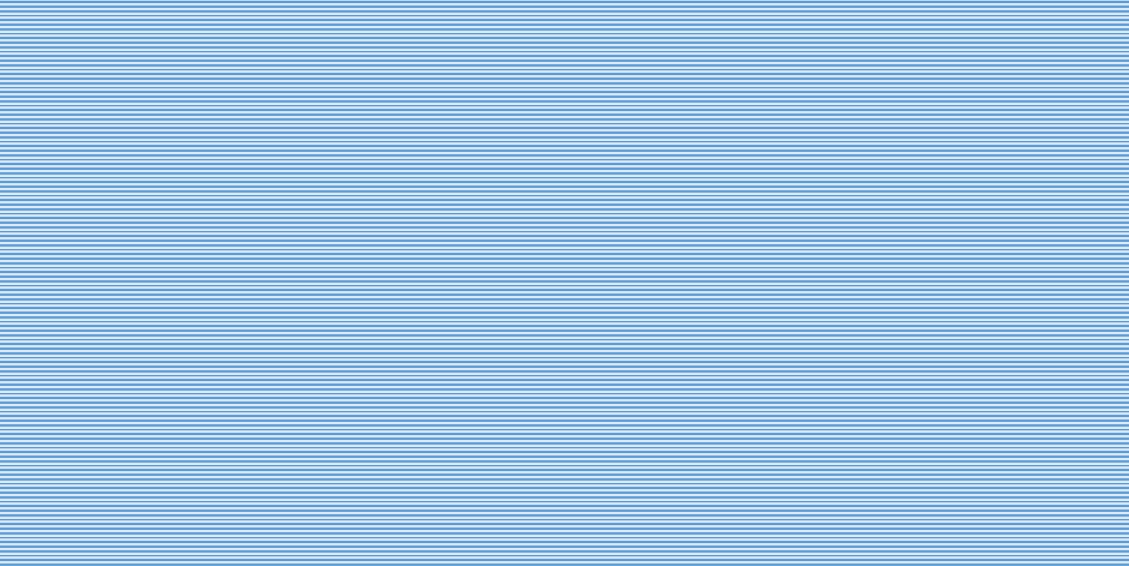

**Figure 5.** The compressive strength value of AC- and ACF-treated soil.

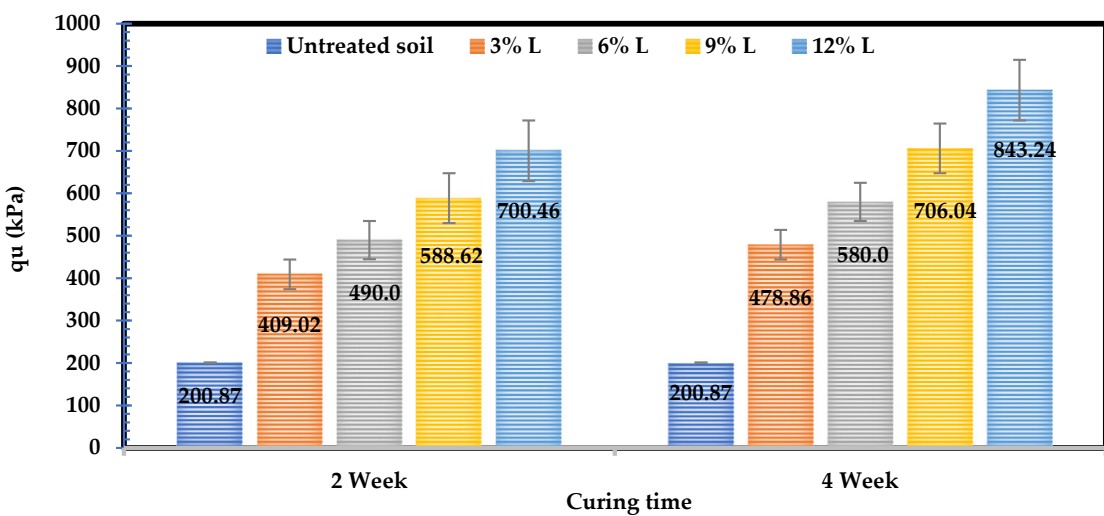

**Figure 6.** The compressive strength value for different lime contents.

The UCS values in lime-stabilised soil improved as the lime percentage and cure time increased [63]. With increasing lime content, the pH value of soil is increased, and it facilitates the dissolving of siliceous and aluminous chemicals from the lattice of clay minerals during lime stabilisation processes. Calcium aluminium hydrate and calcium silicate hydrate gels are formed due to calcium ions combined with compounds released from the lattice of clay minerals [50]. The UCS value for four-week specimens improved by 18.94% rather than two-week samples. Even though the overall UCS values in Figure 6 indicate that lime content and curing time improve compressive strength, the effectiveness of 12 per cent lime is remarkable. A minimum UCS of 800 kPa is acceptable for constructing medium and low-volume roads, according to the Malaysia Public Works Department (MPWD) requirement. The percentages lower than 12 per cent could not enhance the soil strength adequately due to a pH value lower than 12.4 [57]. The result obtained in this study is comparable to previous findings, in which lime in clay soil improved UCS value and was resistant to compression [64].

Figure 7 illustrates that all treated specimens have more strength than untreated soil specimens. It indicates that adding fibre to AC soils enhances soil strength significantly due to fibre promoting interlocking between soil particles and additives [26]. The UCS value in 3% ACF increased by more than 800 kPa, which is the minimum requirement of (MPWD) specifications, while this amount is obtained for 12% of lime after four weeks. Anggraini et al. [65] observed similar results by including lime and coir fibres in the marian clay soil. They reported enhancement in geotechnical properties such as shear strength parameters, flexibility, and UCS values. Wang et al. (2019) discovered that incorporating wheat straw fibres and lime into soil increased strain-softening behaviour, secant modulus, and shear strength [30,65]. Adding fibre as reinforcement improves sample ductility and stability [28]. Therefore, according to MPWD, 3% ACF and 12% L were adequate to obtain a minimal UCS value, as shown in Figure 7.

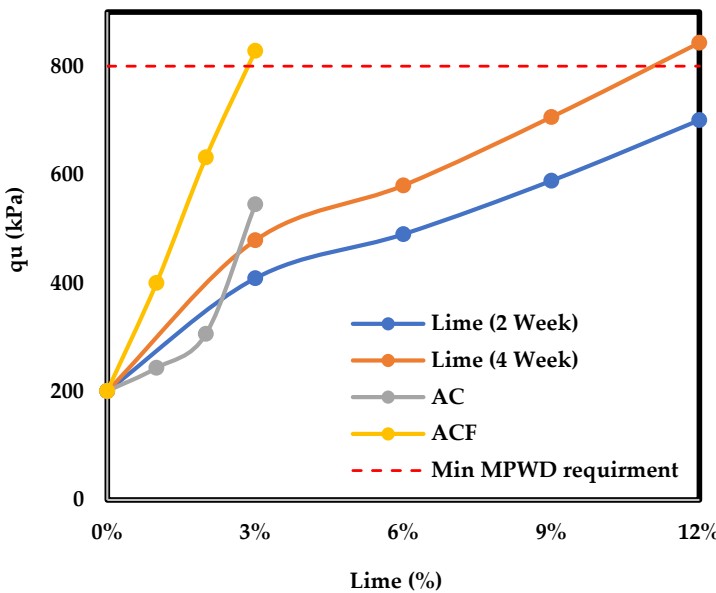

**Figure 7.** The compressive strength value for different lime and AC contents.

### 3.3. Deformability Index ($I_D$) and Elastic Modulus ($E_{50}$)

The influence of ACF on the flexibility and stiffness of soil is investigated with the deformability index. The deformability index is the strain at the peak strength of stabilised soil per strain at the peak strength of original soil. It is a factor utilised to explain the deformation behaviour of soil in this study [66]. This parameter has indicated the deformation behaviour of modified soil compared to unmodified soil [67]. UCS results show $I_D$ rises from 1.02 to 1.36 for 1% AC to 3%ACF. It confirms that fibre and AC improve soil behaviour from brittle to ductile [68]. However, $I_D$ for lime-stabilised specimens is less than one and reduces with increasing lime content, which shows that lime increases brittles in lateritic soil [69].

The influence of AC, ACF, and lime on flexibility and stiffness of soil is also assessed with secant modulus. The secant modulus is regarded as half of the maximum UCS values, as in previous studies [69,70]. Figure 8 shows the elastic modulus values for various combinations. The elastic modulus was raised in AC, ACF, and lime treated with rising content further in 3% ACF and 3% lime; the difference between elastic modulus is insignificant. It is clear that lime increases stiffness in soil due to hydration and pozzolanic processes in lime-treated soil [15].

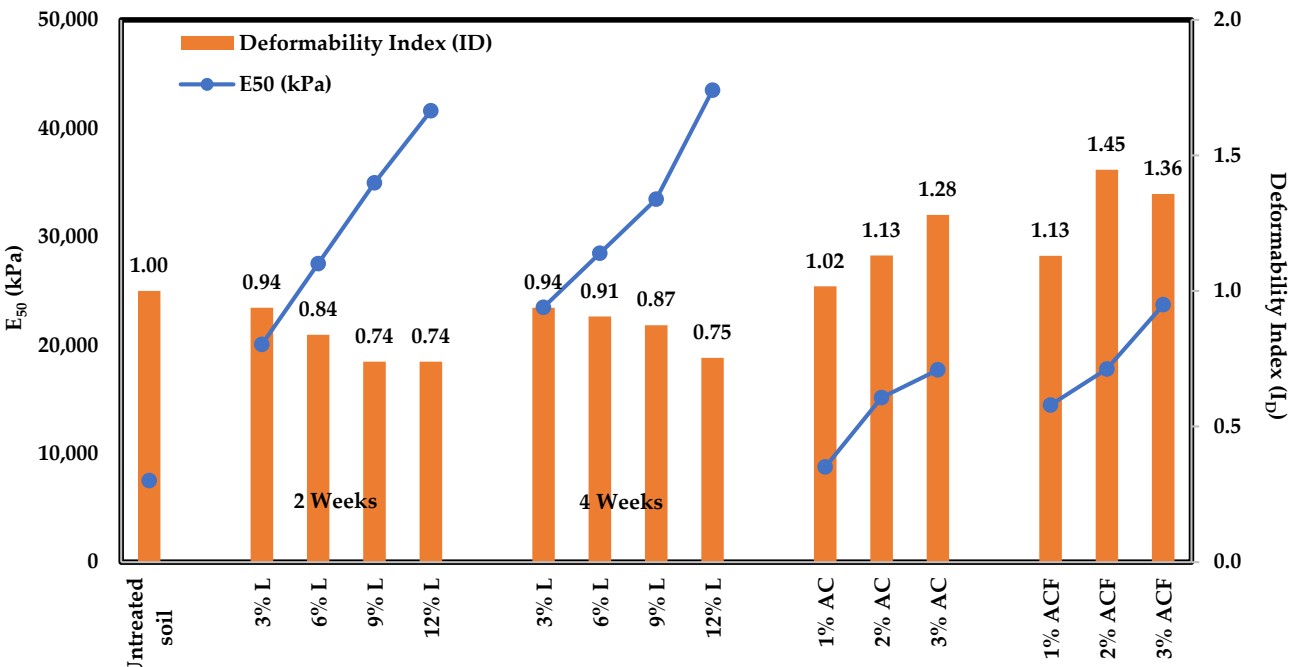

**Figure 8.** Deformability index ($I_D$) and elastic modulus ($E_{50}$) of lateritic soil and treated soil.

### 3.4. One-Dimensional Consolidation

Figure 9 illustrates the e − log P curves and shows that adding 3% ACF and 12% lime has improved the swelling and compression factors. The compression index ($C_C$) has declined by 51% and 78% with the addition of 3% ACF and 12% L, respectively. The interaction of released clay silica and alumina with free additive ions linked the soil particles together in the pore water, improving compressibility. Similar behaviour was seen by [71] when conducting 1-D oedometer tests on soil stabilised lime; lime decreases the void ratio of treated soil. The cementitious gels fill the gaps and voids in the soil structure and significantly reduce the void ratio. Moreover, smaller particles in the AC can minimise the compressibility of parent soil by filling the voids [39].

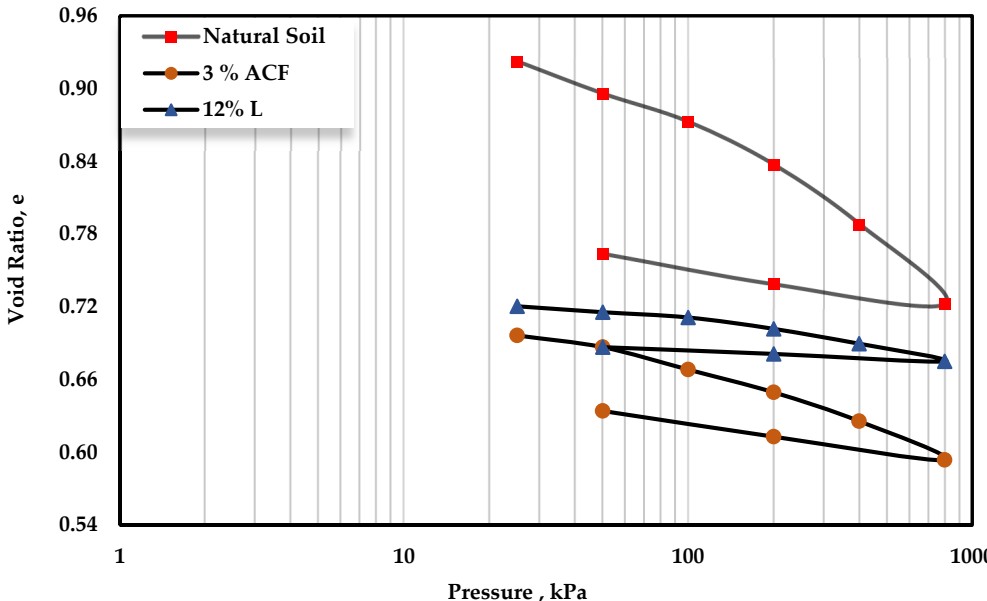

**Figure 9.** Compressibility curves (e − log P).

Figure 10 indicates the void ratio versus consolidation coefficient ($C_v$) for optimum contents and untreated soil. As can be seen, the stabilised specimens have lower void ratios and, hence, less permeability than untreated soil at the same effective vertical stress. This decline in the void ratio of the stabilised specimens is due to AC and lime, which significantly reduces air pockets' volume among soil particles throughout the compaction [72]. Adding 3% ACF reduces the porosity of the non-stabilised soil due to activated carbon is contained small particles with high surface area [73]. However, the $C_v$ reduces in lime-treated soil more than in AC specimens. The reduction in $C_v$ values results from the change in the soil structure due to the hydration and pozzolanic processes between soil particles, resulting in a dense and compact mixture. Hence, the lower the Cv value within the soil, the less permeability was found [74].

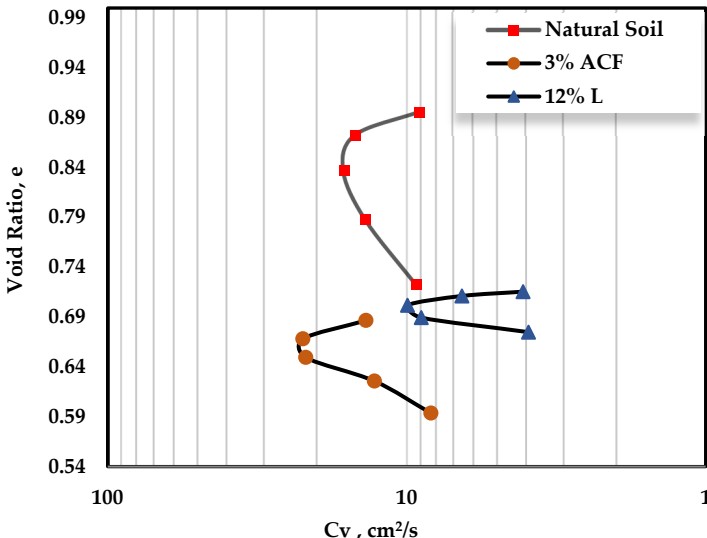

**Figure 10.** Consolidation coefficient.

Figure 11 shows the influence of lime and AC on the permeability coefficient (k) of lateritic soil. The k values for both modified and unmodified soil samples were determined at all stress levels, and specimens were kept saturated in the oedometer cell. However, Mengue et al. [75] recommended calculating the impact of binder content on the permeability coefficient at the highest applied pressure (800 kN/m$^2$). As expected, the permeability of original soil samples is comparatively high ($5.40 \times 10^{-8}$ m/s), whereas that in treated specimens decreases to $2.90 \times 10^{-8}$ m/s and $6.76 \times 10^{-9}$ m/s for 3% ACF and 12% lime, respectively. Hydraulic conductivity is a measure of how easily water can flow through interconnected soil voids. Using materials in highway construction depends on the degree of permeability that is effective in the drainage of the pavement system. In this study, AC fills the voids in the stabilised specimen, restricts water, and enhances bonding with the soil particles. Moreover, the free cations of AC interact with the adsorbed clay mineral cations, causing the diffused water layer covering clay particles to shrink. However, lime fills the voids and gaps with cementation gels during curing time and obstructs water flow [76].

According to Figure 11, the hydraulic conductivity decreases by adding 3% ACF and 12% lime. Compared to untreated soil, the hydraulic conductivity declined about 85% and 26% in lime-treated soil and AC specimens. The hydraulic conductivity would decrease as the curing time in the lime-treated specimen was extended, and more cementitious compounds were produced to fill the voids. Higher lime content causes more cementation bonding, making specimens stiffer and demanding more pressure to reach, yielding strength and starting particle displacement [77].

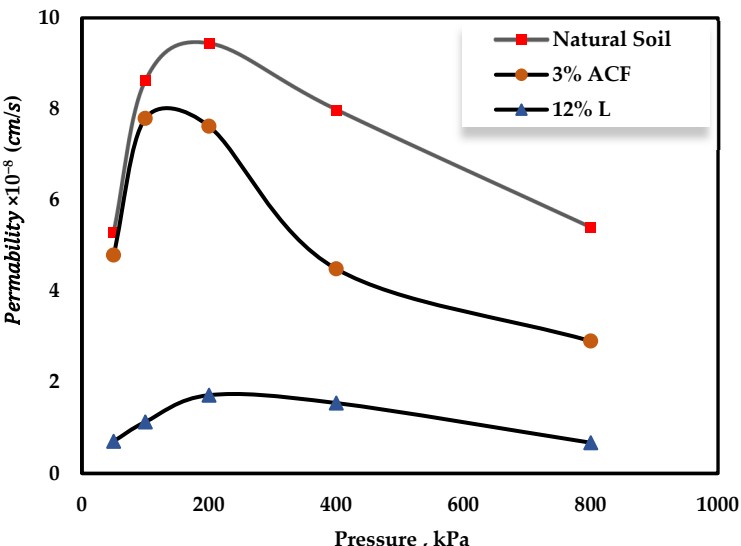

**Figure 11.** Permeability coefficient.

### 3.5. Microstructural Analysis

The FESEM and EDX results in Figure 12 confirmed that the cementitious gels in lime-stabilised specimens decline permeability at the end of four weeks of curing time. Lateritic soil has ample spaces and is more flocculated, whereas larger voids in lime and AC-treated specimens are filled with cementitious gel [78]. These cementitious gels produced are characterised by their low volume change and high strength [79].

The results of the EDX analysis reveal a few changes in the chemical components of the natural and AC stabilised lateritic soils. The lateritic soil contains varying amounts of Al, O, Si, K, and Fe [80]. The 3% ACF treated has C along with Al, O, Si, K, and Fe [80]. Activated carbon appeared in the treated soil with the element calcium carbonate, a cementation agent that reduces the void ratio [81]. It uses physicochemical mechanisms to bind soil particles together, producing a solid soil structure [82]. The lime-treated sample has Ca along with Al, O, Si, K, and Fe. A high calcium weight percentage in lime-treated soil resulted in increased gel production. Thus, the increase in lime content could be attributed to calcium production [79].

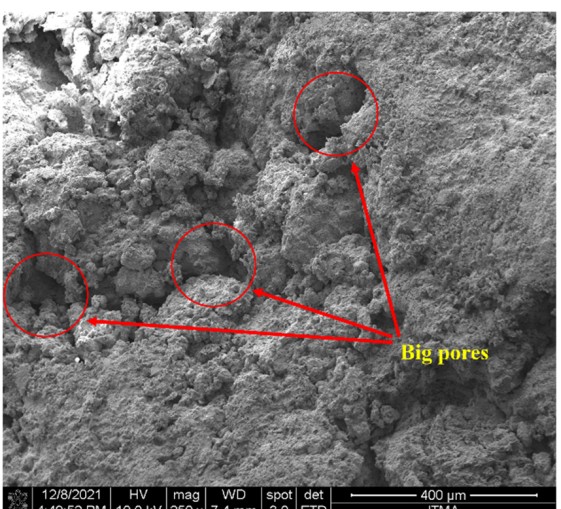

| Element | O | Al | Si | Fe | K |
|---|---|---|---|---|---|
| Weight (%) | 54.54 | 16.29 | 15.54 | 12.74 | 0.89 |
| Atomic (%) | 70.79 | 12.04 | 70.79 | 4.74 | 0.74 |

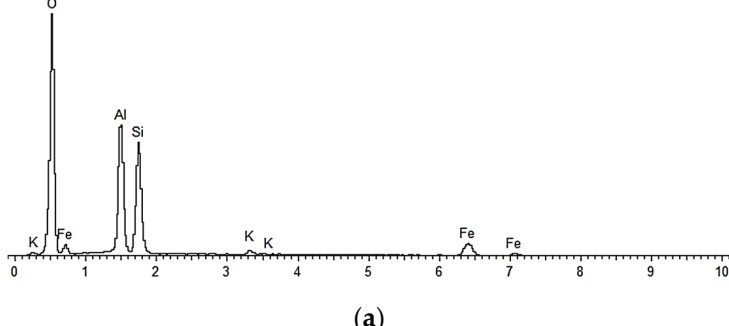

(**a**)

**Figure 12.** *Cont.*

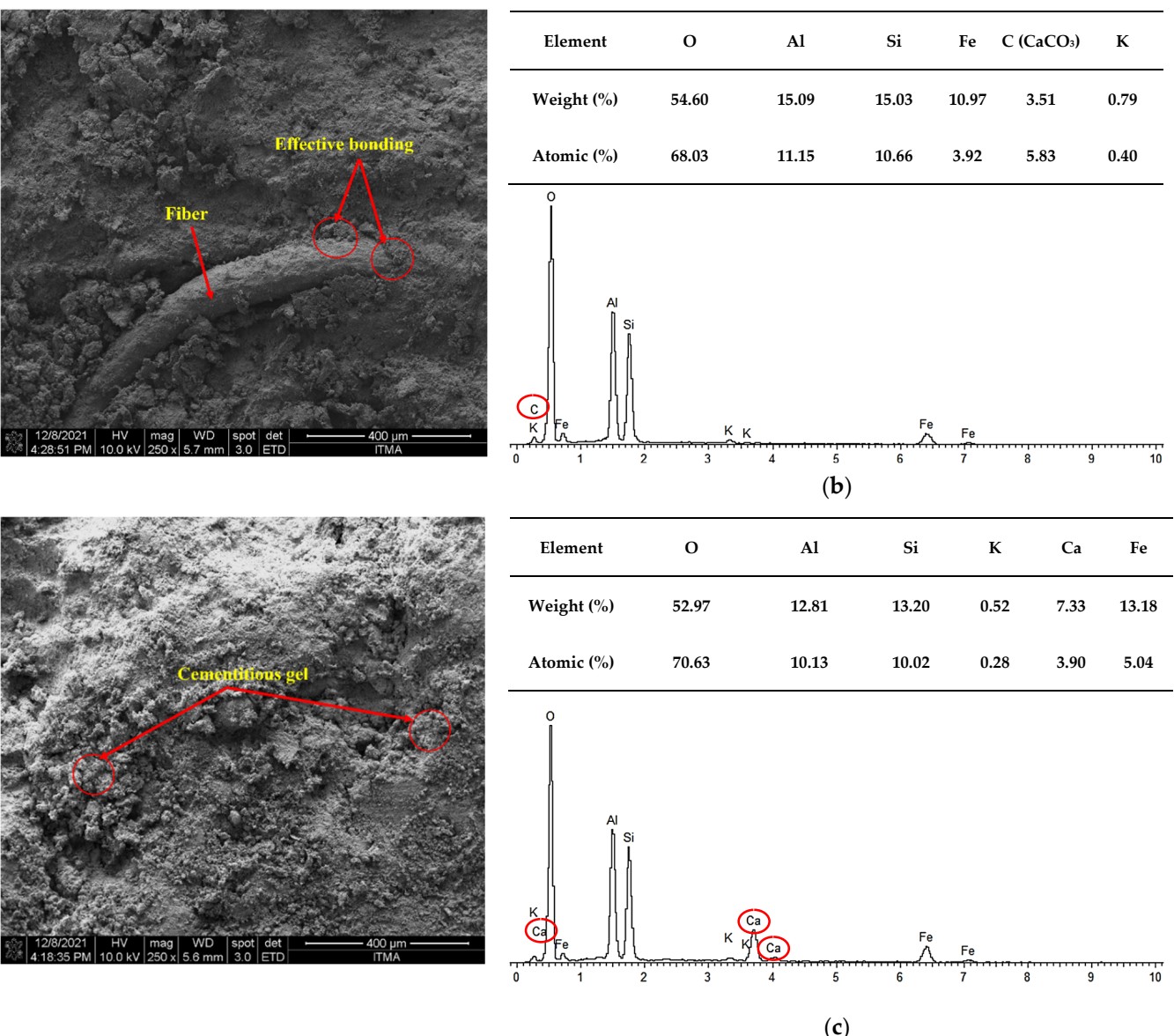

**Figure 12.** FESEM images of (**a**) lateritic soil, (**b**) AC specimen, and (**c**) lime specimen.

### 3.6. Surface Area Analysis

The BET surface area technique was used to assess the changes in surface area and micropores of untreated soil, 12% lime, and 3% ACF-treated soil. The impact of lime and AC on the surface area, pore volume, and pore size are shown in Figure 13. The BET values of the unmodified soil and 3% ACF increased from 25.57 m$^2$/g to 55.57 m$^2$/g, while pore size and pore volume decreased from $3.05 \times 10^{-7}$dm and $0.389 \times 10^{-3}\frac{dm^3}{g}$ to $7.42 \times 10^{-8}$dm and $0.330 \times 10^{-3}\frac{dm^3}{g}$, respectively. Although the BET value for 12% lime increased to 28.07 m$^2$/g, it is insignificant compared to 3% ACF. The pore size and volume increased considerably in comparison with 3% ACF. The BET results confirmed that 3% ACF is more effective than 12% lime in terms of surface area and BET value. To better understand the effect of AC in decreasing pore volume and pore size, the BET result of 3%L-treated soil was added. Therefore, activated carbon improves the lateritic soil structure into a completely interlocking system with fewer tiny pores and prepares the situation for more reactions by raising the surface area [73].

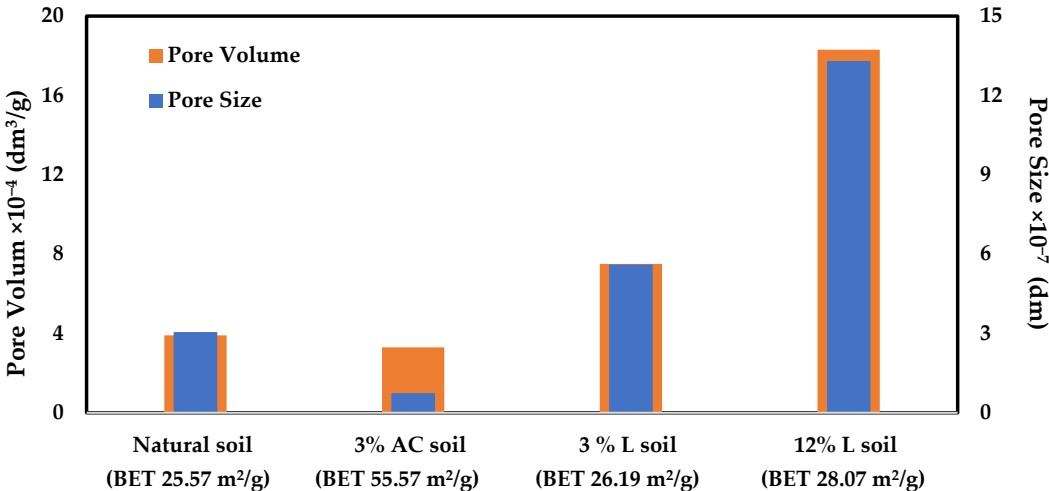

**Figure 13.** Pore volume, surface area, and pore size.

## 4. Conclusions

This study evaluates lime, AC, and ACF stabilised subgrade soils through consolidation tests, UCS tests, and microstructural analyses. According to MPWD standards, the findings showed that lime and ACF enhance the performance of this lateritic soil as a subgrade in road construction. Based on the result of the experiment work, 12% lime is the lowest content of lime that should be included in the soil to enhance the strength of soil significantly because, in this study, the soil is acidic, and a high percentage of lime is needed to reach pH value 12.4. Moreover, the UCS values confirmed that 12% lime is optimum content based on Malaysian standards. Although lime content and curing age affect UCS values of lime treatment, 3% ACF has the same UCS value as 12% lime (four weeks age). In addition, the elastic module increases with increasing additives content in lime-treated and ACF-treated soil. The elastic modulus of 3% ACF and 3% lime are almost identical when comparing lime and ACF specimens. Using lime and ACF effectively enhances the consolidation coefficient in stabilised soil due to decreasing the void ratio and hydraulic conductivity by them. However, $I_D$ decreased in lime-treated soil with rising lime content and curing time, while $I_D$ increased in AC and ACF-treated soil with growing AC content. The BET results confirmed that activated carbon improves BET value significantly compared to lime. Hence, ACF treatment can improve the compression strength of lateritic soils. Still, it needs more investigation to determine the influence of ACF on other strength factors under various loading and environmental conditions.

**Author Contributions:** The conceptualisation, design of the work to data gathering, data analysis, manuscript writing, and review have done by contributions of S.T., N.N.N.D., F.M.J., F.M.K., A.S.A.R. and M.J.R. to the preparation of this paper. All authors have read and agreed to the published version of the manuscript.

**Funding:** This study was funded by Universiti Putra Malaysia under project code: GP-IPS/2021/9701400, grant type: Geran Inisiatif Putra Siswazah (GP-IPS).

**Institutional Review Board Statement:** Not applicable.

**Informed Consent Statement:** Not applicable.

**Data Availability Statement:** The data used in this research can be provided by the corresponding author upon request. Because of privacy concerns, the data is not publicly available.

**Acknowledgments:** The authors are thankful to Universiti Putra Malaysia for their cooperation in launching this research.

**Conflicts of Interest:** The authors declare that this research presented contains no commercial or associative interests that would create a conflict of interest.

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
