# Peer review of "Performance Evaluation of Lateritic Subgrade Soil Treated with Lime and Coir Fibre-Activated Carbon"

_applsci, doi:10.3390/app12168279_

Round 1
Reviewer 1 Report
This study investigates the influence of lime and activated carbon with coir fibre (ACF) as waste materials and an environmentally friendly binder to stabilise lateritic soils. Experiments, including the one-dimensional consolidation and unconfined compressive strength (UCS) tests, have been conducted to investigate the geotechnical properties of stabilised soil in various percentages of additives 3%,6%,9%, and 12% lime and 1%, 2%, and 3% ACF. The study is well writer but needs a revision based on following comments:
Abstract: The authors need to revise the abstract. Highlight the scientific value added by your paper in your abstract. The abstract should clearly describe the core of the problem you are addressing, what you did, found, and recommended to the readers. It will help prospective readers of the abstract to decide if they wish to read the entire article.
Lines 66-67, authors need to discuss the chemical composition of Activated Carbon & Coir fibre and the amount of AC and CF generation each year globally.
Figures 5, 6 need error bars.
The conclusion section needs to be re-written by including the important results, analysis and validation outcomes in points or bullet format. The optimum dosages of wastes in the soil need to be provided in the conclusion and abstract.
Author Response
Thanks for the comments. All responds appeared in attached file.

Reviewer 2 Report
The paper investigates the influence of lime and activated carbon with coir fiber (ACF) as waste materials to stabilize lateritic soils. The methodology mainly concentrates on performing experiments of one-dimensional consolidation and unconfined compressive strength (UCS) tests to investigate the geotechnical properties of stabilized soil in various percentages of additives 3%,6%,9%, and 12% lime and 1%, 2%, and 3% ACF. The results demonstrate that 3% ACF and 12% lime can significantly improve the strength parameters and decrease the void ratio and permeability in the stabilized soil. Furthermore, microstructural analysis was performed before and after stabilization for optimum content. Both the experimental and microstructural analyses show similar results, and the results were presented in high quality. The reviewer is only concerned about one thing, the increase of lime percentage added to the soil sample. It is obvious that the author adding more lime will cause a more compressive strength increase because the lime strength increases with time as well. But that kind of increase is mainly due to the lime other than the soil property change. So please explain this phenomenon because it is important for this research goal. In summary, the reviewer recommends that the authors can clarify the following issues:
- In Page 3, line 84, there should be a typo “AASSHTO.” The code name is AASHTO. Please specify which edition the author used (7th or 8th) and add a reference to this code. On the other hand, why did the authors decide to use AASHTO to define the soil type other than IBC 2018 or other related codes? Please explain.
- In Page 3, Table 2. Please do not divide a whole Table into two separate pages, except it can not be finished in one full page.
- In Page 3, line 98. Please specify what XRF means. Because the author never explained it before. Please also check the same problem that appeared before, for example, SEM picture (line 58), EDX, BET, and FESEM (line 77). All the abbreviations should be explained the first time they appear.
- In Page 4, line 121. Please specify the method “BS1377: Part 7: 1990” comes from which code/standard (also for line 126). Please also add a reference (link) for this method (BS1377).
- In Page 5, why did the authors test the pH value of the soil sample? Is there any help for this paper’s main research goal?
- In Page 6, Figure 4’s title. There should be a typo. Where the Figure 4(d) comes from? Please revise it.
- In Page 7, Figure 6. The reviewer thinks that the compressive increase by adding more lime is mainly due to the lime strength increase (like concrete) other than the soil property change. How do the authors clarify this phenomenon? Please explain.
- In Page 11, Figure 12, what does “Big porc” mean? The resolution of this Figure is lower than the other Figures shown in this paper. Please replot the raw data if the authors have the raw data in Microsoft Excel. Please add annotation by Microsoft software as well. In this way, it will have high resolution.
Finally, the reviewer will give two suggestions for the authors. (1) Please make sure all the words shown in all Tables in this paper have the same font type and size. (2) Please check the word spelling and typos shown in this manuscript.
Author Response

(The authors gave the same response as above.)
